# Secondary Prevention of Depressive Prodrome in Adolescents: Before and after Attending a Jogging Program on Campus

**DOI:** 10.3390/ijerph17217705

**Published:** 2020-10-22

**Authors:** Ke Tien Yen, Shen Cherng

**Affiliations:** 1Department of Leisure and Sports Management, Chengshiu University, Kaohsiung 83347, Taiwan; ktyen2006@gmail.com; 2Center for Environmental Toxin and Emerging-Contaminant Research, Chengshiu University, Kaohsiung 83347, Taiwan; 3Department of Computer Science and Information Engineering, Chengshiu University, Kaohsiung 83347, Taiwan

**Keywords:** depressive prodrome, subthreshold depression, secondary prevention, primary care, logistic model, false positive rate, DSM-5, depression

## Abstract

The adolescent depressive prodrome has been conceptualized as an early integrated sign of depressive symptoms, which may develop to a first episode of depression or return to normal for the adolescents. In this study, depressive prodrome presented the early self-rated depressive symptoms for the sample participants. By referring to the Kutcher Adolescent Depression Scale and the psychometric characteristics of the Adolescent Depression Scale (ADR), we proposed a self-rated questionnaire to assess the severity of the depressive symptoms in adolescents before and after attending the jogging program on a high school campus in Taiwan. With the parental co-signature and self-signed informed consent form, 284 high school students under the average age of 15 years, participated in this study in March 2019. Through the software of IBMSPSS 25, we used a binary logistic model, principal component analysis (PCA), multiple-dimensional analysis, and receiver operating characteristic curve (ROC) to analyze the severity of the depressive prodrome via the threshold severity score (SC) and false positive rate (FPR). Findings revealed that attending the 15-week jogging program (3 times a week, 45 min each) on campus can change the severity status and reduce the prevalence of moderate-severe depressive prodrome by 26%. The two-dimensional approach identified three symptoms, which were the crying spell, loss of pleasure doing daily activities, and feeling the decline in memory. They kept being invariant symptoms during the course of depressive prodrome assessment for sample participants. In this study, the campus jogging program appeared to be able to affect the FPR of the measure of depressive prodrome. Compared with the subthreshold depression, the depressive prodrome emphasized the assessment from the view of the secondary prevention by representing the change from a person’s premorbid functioning up until the first onset of depression or returning to normal. However, the subthreshold depression is a form of minor depression according to DSM-5 criteria varying on the number of symptoms and duration required, highly prevalent in the concern of primary care.

## 1. Introduction

Adolescent depression broadly describes the emotional state, syndrome, and a group of mental disorders with a cluster of specific symptoms and associated impairment [1,2,3] in adolescents. It becomes an epidemic concern globally, with the prevalence rate of 13.3% in the United States and 8.9% among people aged 15–18 in Taiwan [4,5,6]. On the basis that 2.5 million children and adolescents attended schools in Taiwan, the prevalence of mental health problems was about 0.2 million children and adolescents requiring treatment in the year of 2019 [7]. To date, the factor increasing the risk for developing mental health problems is still a major concern [8] in high school, especially adolescent depression appears in different forms that may not be formalized in DSM-5 [9]. Adolescent depression ranges from mild to severe, transient to chronic or recurrent [10], and multifactor with a combination of genetic, biological, environmental, and psychological factors in etiology [11]. For the purpose of secondary prevention of depressive first episode in adolescents, we concern about how we know what adolescent students are like in terms of depressive symptoms before the onset of depression so that we can trigger the intervention. Moreover, we also concern about the changes in depressive symptoms from prodrome to the onset or returning to normal. Referred to the criteria in DSM-5, depression must experience five or more symptoms during the same 2-week period and at least one of the symptoms should be either (1) depressed mood or (2) anhedonia (loss of interest or pleasure) [12]. We recognized that the criteria to identify adolescent depression are no different from the adult one [13]. As the adolescent depression arises from the exposure of young people to specific risk factors of low self-esteem, gender discrimination, negative body image, less social support, negative cognitive style, low coping ability and nonspecific risk factors of poverty, violence, social isolation, child abuse, and family breakdown, etcetera [14], the definitive diagnosis of adolescent depression requires the fulfillment of criteria in terms of symptoms, severity, and duration of disturbance presented in DSM-5 [15,16]. However, if the early signs of the depressive symptoms of the adolescent depression appear but not yet clinically specific or severe, alternatively, we define it as an adolescent depressive prodrome (ADP) [17]. The ADP can be classified into three types, including APS (attenuated positive symptom prodromal syndrome), BIPS (brief intermittent prodromal syndrome), and GRDS (deterioration prodromal syndrome) [18]. At this time, we limit our study of the state of ADP identified by the BIPS before the onset of depressive-episode to make reliable estimates of depressive symptoms in adolescents [19,20]. Previous research illustrated that conditions of ADP might last for a considerable duration and cross the threshold to manifest the clinical depression [15]. It is likely to define the ADP as an early marker of adolescent depression (AD) through the criteria of symptoms and signs defined in DSM-5 [17]. Basically, ADP with early symptoms that have passed an uncertain period will be milder than the clinical stage of AD [21]. It is more likely to be classified into the catalog of mild adolescent depression with a period of discomfort, which should be the very early stage of the first episode of depression for adolescents [22] and occurs before the full diagnostic features appear [23,24,25,26,27]. In this study, we conceptualized the threshold score to describe the prevalence, false positive rate, sensitivity, and reliability [28,29] of the manifestation of the depressive symptoms. Usually, depression screening uses a preset threshold to identify the severity of state status [19]. Previous articles reported several self-rated measures to identify the depressive symptoms with full confidence [19,25,30,31] but no clinical interviews being referred to support the diagnosis, the rate of misjudgment was quite high [32,33]. On the base of the Kutcher Adolescent Depression Scale, Beck Depression Inventory (BDI), and Hamilton rating scale for depression [3,34,35,36], we established a self-rate measure to assess the adolescent depressive prodrome [36,37], which evolves 16 depressive symptoms defined by DSM-5 with a 4-point scale and ranged total manifestation scores from 16–64 [38,39]. As false positive rate (FPR) is the probability of falsely rejecting the null hypothesis of a particular test, and sensitivity measures the proportion of true positives that are correctly identified [40]; when developing a screening instrument, we have to face with the balance between sensitivity and FPR [26,41]. To illustrate the performance of our self-rated measure, we used the preset threshold of severity score (SC) via the ROC analysis and binary logistic model to examine the sensitivity and FPR of the assessment of ADP [42]. Reducing the FPR to avoid the high rate of misjudgments [17,43,44,45] is a challenge [46] to identify the ADP. In this study, we highlighted the attending campus jogging program as an intervention for the students to reduce FPR of the assessment of ADP. Through the parameterized SC of the manifest symptoms to describe the status of the severity of depressive prodrome [47,48,49], we specified the underlying factors from the self-rated measure, which are usually difficult to be recognized [50,51]. Previously published research papers have shown that exercise functions not only as an intervention but also as a method help people recognize the precursor when the onset of depressive disorders cannot be specified [52,53]. We proposed a campus jogging program for a 15-week period, 3 times a week according to the planned date, each time with 10 min of warm-up exercise and 35 min of jogging as a preventive intervention for the participants in this research. Three counselors and a coach followed with the students for the safety concern when jogging on campus and students were free to chat with peers, coaches, and counselors. As a strong association with the all-cause risk for mental health [54], jogging is universal level prevention (secondary prevention) for depression [55]. From the research perspective, the dimensional approach from self-rated measure along with DSM’s set of symptoms [56] allows us more latitude to assess the ADP. Its inclusion provides more utility in research contexts. However, students should not be treated as patients, and we modified the form of assessment measure by Section III in DSM-5. We kept 12 domains of parent/guardian-rated level 1 cross-cutting symptom measure but added more questions and used a 4-point Likert scale for the assessment of ADP based upon the Kutcher Adolescent Depression Scale. Comparatively, although the DSM-5 Section III [57] provided a substantially different taxonomic structure for depression, the associations between the scale of DSM-5 Section III and our dimensional approach of depressive prodrome in adolescents render the depressive symptoms in combination with DSM’s set of core symptoms [58]. Similarly, the feature of subthreshold is widely used in medicine to label individuals who are in the early stages of a disease process and to identify high-risk populations that need to be monitored or provide with specific interventions or treatments. Though, subthreshold depression should refer to an individual who has not previously met the full criteria for major depression but currently experiences depressive symptoms that are not severe enough or persistent enough to merit a diagnosis of major depression. We believe that appropriate treatment of subthreshold depression is necessary for primary care. Therefore, it should not be regarded as a preventive stage for secondary prevention of prodromal depressive symptoms in adolescents.

## 2. Methods

### 2.1. Sample Participants

In March 2019, 284 Taiwanese adolescent students in a public high school participated in this study. All the participants were required to submit the parental cosigned and self-signed informed consent form. The university ethic committee approval of the study was granted on the day of July 31, 2018 (Approval No.: Research 0012E-2018-2020). The self-rated measure was used to evaluate the prevalence and FPR of the assessment of adolescent depressive prodrome (ADP) for the sample population. The university ethic committee approved this study to use the data of self-rated measure for publishing research results only. All the data should not be used for any clinical consideration. Professional assistants explained the procedures and objectives of the study to the students and interpreted the symptoms stated in the self-rated measure to the participants clearly. It only took less than 30 min for the participants to answer the questionnaire. Meanwhile, we encouraged the participants to make appointments with the professional counselors asking questions about the concerns of secondary prevention of the adolescent depression on campus [59,60].

### 2.2. Modeling Procedures

The receiver operating characteristic (ROC) curve describes the relative change between FPR (false positive rate) and TPR (true positive rate) for the classification confusion matrix [61]. In this study, we used the ROC curve to determine the threshold from the manifest score of the self-rated measure to assess the various states of the severity of depressive prodrome [62]. The proposed self-rated measure was a construct of 16 questionings of depressive symptoms of the depressive prodrome for the participants. Each symptom was self-rated on a 4-point Likert scale question to describe the severity of the symptoms (1—asymptomatic, 2—symptom appears one or two times a week as mild, 3—symptom appears three times a week as moderate, and 4—symptom appears more than three times a week as severe). In Table 1, we listed questioning of the symptoms noted as Q1, Q2…, and Q16, the Cronbach alpha coefficient of the questionnaire and the frequency of self-scoring of each symptom [63] for the participants. In Table 2, we listed the R-squared of the measure to demonstrate the proportion of the variance for the depressive prodrome being explained by symptoms described as Q1, Q2…, and Q16. Using Principal Component Analysis (PCA), we extracted the three latent factors BF1, BF2, and BF3 before attending the jogging program and AF1, AF2, and AF3 after attending the jogging program and listed the analysis results in Table 2 and Table 3. These factors contained one emotion factor and two non-emotional factors [64]. Since most of the prodromal depressive symptoms were associated with emotional factors only, we took the cutoff factor score of the emotional factor at 2.5 points [15]. When the factor score was less than the 2.5, we coded it 0; however, if it was greater than or equal to the 2.5, we coded it 1, which then we converted the continuous factor score to a binary set of data. The ROC curve analysis with the thresholds of the measure is shown in Figure 1. Following with presetting the threshold, we constructed a binary logistic regression model to calculate the predictive probability of the occurrence, sensitivity, and FPR of the event of depressive prodrome in adolescents [65]. The observed false positive rates, sensitivity, specificity, false negative rates, predicted probability, and prevalence [66] were shown in Table 3. We showed the ROC analysis with the thresholds of the measure in Figure 1. The results of the dimensional scale analysis were then shown in Figure 2, Table 4 and Table 5. On the other hand, the prevalence vs. the FPR with the states of the severity of the ADP is shown in Figure 3. By reviewing the relevant literature, we learned that latent factors of the self-rated measure of depressive symptoms can be evolved by a two-dimensional approach to emotional and non-emotional factors [67,68]. From Figure 3, we observed that jogging significantly affects the severity status of the depressive symptoms in adolescents [69]. The severity scores (SC) of four thresholds at 35, 36, 37, and 38 of the measure vs. PR and FPR were recognized from ROC analysis in this study. Figure 1A shows the threshold level of depressive prodrome as SC, where SC = 35 described the light, SC = 36 described the mild, SC = 37 described moderate, SC = 38 described severely, and SC = 39 described very severely of the depressive prodrome [70]. In Figure 2, via the two-dimensional scale analysis, we disclosed the association of the symptoms between each other for both before and after attending the jogging class.

## 3. Results and Discussion

In this study, 284 participants participated, i.e., 140 were girls (49%) and 144 were boys (51%). The average age of the participants was 15 years and they were all high school students. The ADP status can be described by latent factors recognized [71,72] from the PCA analysis and shown in Equations (1) and (2). Equation (1) described the ADP status after the intervention,
|ADP) _A_ = 0.70|AF1) + 0.15|AF2) + 0.15|AF3).(1)

In Equation (1), ADP was 70% characterized by AF1, which was a comorbid state of eight integrated symptoms and shown in Table 2. Meanwhile, in Equation (2), ADP status before the intervention was 74% characterized by BF1.
|ADP) _B_ = 0.74|BF1) + 0.14|BF2) + 0.12|BF3).(2)

Via the preventive intervention, the most dominant symptom altered from Q13 to Q1 and Q2. We observed the ADP is like a common cold, comes and goes [73] with little warning signals. From Table 4, we used the multidimensional scale (MS) approach to reveal the similarity. Before attending the jogging program, we preset the MS distance at 0.8 for the similarity analysis of 16 symptoms. The analysis result disclosed the symptoms of self-loathing, irritability, sleep disturbance, appetite change, unexplained aches, and pains, feeling nervousness, feeling fatigue, loss of interest in daily activities, feeling short of attention, slowing down of thought, and feeling hopelessness were comorbid and highly similar to crying spells. However, the symptom of loss confidence was comorbid to sickness. The symptoms that can be clearly distinguished with less similarity were crying spells, loss of pleasure, feeling the decline in memory, loss of confidence, and having pessimism comforting delusion. Consequently, five symptoms, Q1, Q9, Q10, Q13, and Q14 should be enough to reveal the depressive prodrome by the self-rated measure at the SC = 36 threshold. This result is consistent with the diagnostic requirements of DSM-5 with a minimum of five depressive symptoms. Nevertheless, the prodromal symptoms we defined in this study were all based on the diagnostic criteria of DSM-5 except for the restriction of the duration of symptoms in the course of the prodrome. After attending the campus jogging program, the similarity analysis with presetting of MS distance 0.8 for the sample participants disclosed that self-loathing, irritability, sleep disturbance, appetite change, unexplained aches and pains, feeling nervousness, feeling short of attention, lose confidence, and pessimism comforting with delusion, and sickness with fever were highly similar to crying spells. The Q1 would be then comorbid with Q8, Q9, Q10, Q12, and Q15 to reveal symptoms that persist after the intervention. In Table 3, we revealed the FPR, sensitivity, specificity, and prevalence at the different thresholds (presented by SC values) of the manifest score of the self-rated measure before and after attending the campus jogging program.

For the sample participants, attending campus jogging program significantly affected the prevalence of depressive prodrome. Previous literatures reported the rate of misdiagnosed depression in adolescents is higher than that in adults [65,74,75], which is consistent with the results of our analysis. We found that attending the campus jogging as an early preventive program significantly increased the FPR of the assessment of depressive prodrome even if it reduced the rate of the prevalence in adolescents. Interestingly, MS revealed the similarity integration of the 16 symptoms into several basic symptoms. The jogging caused the changing of self-rated measure for symptoms from “Q1, Q9, Q10, Q13, Q14” to “Q1, Q8, Q9, Q10, Q12, Q15.” Furthermore, the effect of engagement on the validity of attending the jogging class revealed that during the period of 15 weeks, the number of student counseling appointments increased by 60% compared to the previous semester while the student academic-related issues increased by 40%, on dating issues increased by 20%, and the cases of misconduct decreased by 40%. This study revealed the association of attending jogging program with mental health in high school. Through counseling information, we also assured that campus jogging program is associated with academic achievement and emotional status [76]. Further studies of mental health-enhancing physical activity need to be recognized as an important element for high school students. Nevertheless, jogging is a significant contribution to either clinical or nonclinical level of mental health in adolescents. In comparison to our study with the reference [77], we learned that not only an identified clinical and nonclinical class of depressive symptoms but also non-identified depressive prodrome is vulnerable via the dimensional approach from the self-rated measure. However, we should be careful about the recognition between the subthreshold depression (SD) [78,79] and the depressive prodrome in adolescents. SD has a higher risk of developing persistent depression or major depressive disorder, but the prodrome we studied is only a signal that is used to predict the severity of the early state of depressive symptoms. It may return to normal or have a possibility to develop into depression.

## 4. Conclusions

In this study, the finding specifically revealed that attending a jogging can be considered as an early preventive program to affect the severity of depressive prodrome for the sample participants. Following with attending a 15-week jogging program, the assessment of depressive prodrome via the self-rated measure has shown the variance of a false positive rate due to the transition of the severity state of depressive prodrome. The condition of the same prevalence showed that jogging intervention altered the severity state described by the threshold from moderate (SC = 36) to mild (SC = 35). A 26% decrease in the prevalence (at SC = 36) was revealed in this study. Our analysis results also showed that a higher threshold of the manifest score of the measure means the higher possibility of the misdiagnosis of the severe severity state of depressive prodrome. The association of FPR with the prevalence would be consistent with the previous study [80]. We disclosed that depressive symptoms may not necessarily be distinct from intervention. Our MS analysis provided the strong support that none of the specific symptoms except crying spell (Q1), loss of pleasure doing daily activities (Q9), and feeling the decline in memory (Q10) may be kept invariant during the prodrome course to both the first-episode depression and the period of the depressive prodrome transition to normal. The professionals should carefully evaluate the similarities of the symptoms varied in comorbidity. Secondary prevention of adolescent depression programs [8] should be implemented necessarily in high schools for the concern of adolescent mental health. Furthermore, we’ll expand studies, especially for the Taiwanese youth. Studies include designing the clinical trials to confirm the minimum false positive rate of the dimensional approach to assess the adolescent depression for high school students. Clinical development of depressive prodrome to depression and severity status track of multidimensional approach of continuity of depressive symptoms between clinical and nonclinical levels in adolescent for the population in high schools in Taiwan and understanding of the existing epidemiological symptom threshold for the development of depression should be the must for our advance researches.

## Figures and Tables

**Figure 1 ijerph-17-07705-f001:**
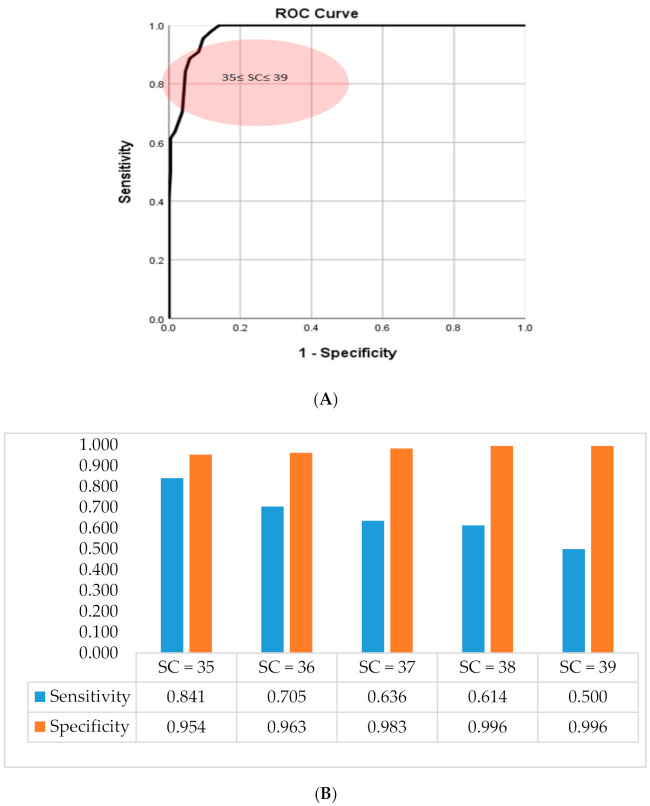
(**A**) The result of receiver operating characteristic curve (ROC) analysis shows the sensitivity (true positive) vs. 1-specificity (false positive rate) with the threshold severity score (SC) of the measure on the condition of cutoff at 2.5 for the emotional latent factor. The area under the curve is 0.978 ± 0.007 between the lower bound of 0.964 and the upper bound0.992) with asymptotic significance 0.000 (**B**) It shows the association of threshold with sensitivity and the specificity by logistic regression model analysis and SC = severity score.

**Figure 2 ijerph-17-07705-f002:**
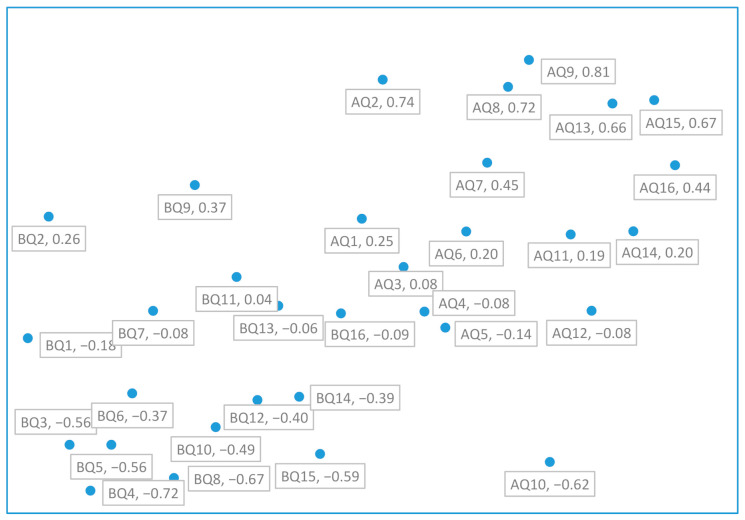
Proximities analysis, (AQ#) means the symptoms with intervention and the (BQ#) means the symptoms without intervention. The distances of every two points mean the similarity between these two symptoms. The longer the distance, the less similar they are, where # means that the numbers ranged from 1 to 16. Meanwhile, (●Q1) represents of crying spells, (●Q2) self-loathing, (●Q3) irritability, (●Q4) sleep disturbance, (●Q5) appetite change, (●Q6) unexplained aches and pains, (●Q7) feeling nervousness, (●Q8) feeling fatigue and loss of interest in daily activities, (●Q9) decrease in feeling joy, (●Q10) feeling decline in memory, (●Q11) feeling short of attention, (●Q12) slowing down of thought, (●Q13) loss confidence, (●Q14) pessimism comforting with delusion, (●Q15) feeling hopeless, and (●Q16) sickness with fever, vomiting, or diarrhea, where “●” can be either A (with intervention) or B (without intervention).

**Figure 3 ijerph-17-07705-f003:**
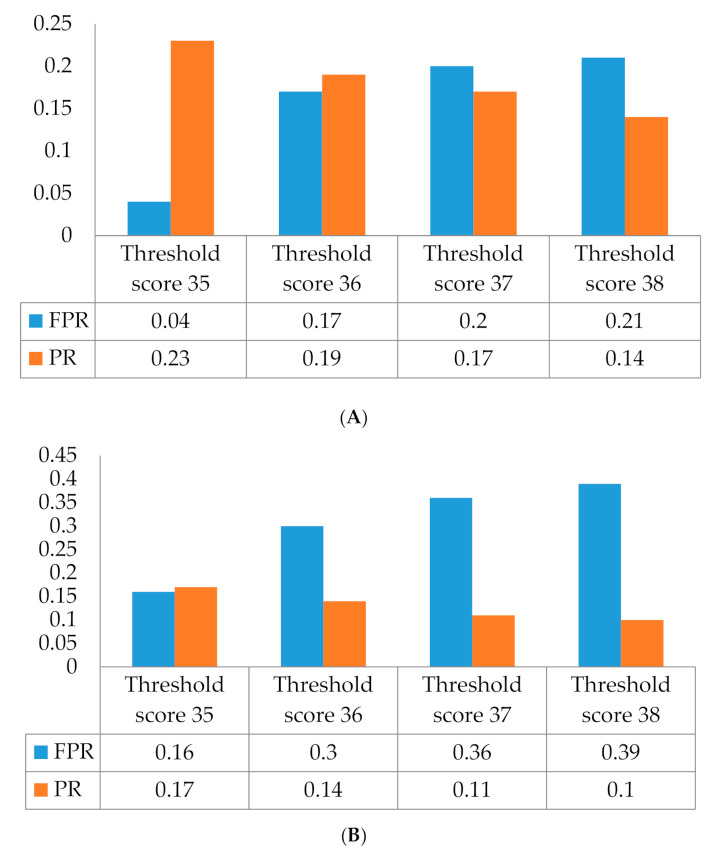
(**A**) It shows the association of the false positive rate (FPR) and prevalence rate (PR) on the threshold score without jogging intervention by the binary logistic model on the condition of cutoff 2.5 for factor score estimate. (**B**) It shows the association of the FPR and the PR at different threshold scores with jogging intervention by logistic model on the condition of cutoff 2.5 for emotional factor score estimate.

**Table 1 ijerph-17-07705-t001:** Reliability Cronbach’s alpha coefficients and frequency for each symptom of the self-rated measure listed.

Index	Inquiry	^†^ Cronbach’s Alpha		Frequency *N* (%)							
				No Risk		Mild		Moderate		High	
		* A	** B	* A	** B	* A	** B	* A	** B	* A	** B
Q1	I have crying spells	0.9	0.91	184 (65)	184 (65)	79 (28)	68 (24)	18 (6)	26 (9)	3 (1)	6 (2)
Q2	I felt blue and depressed			120 (42)	120 (42)	108 (38)	108 (38)	42 (15)	42 (15)	14 (5)	14 (5)
Q3	I feel more likely than ever to lose my temper			192 (68)	192 (68)	73 (26)	66 (23)	16 (5)	20 (7)	3 (1)	6 (2)
Q4	I cannot sleep well			178 (63)	178 (63)	72 (25)	72 (25)	29 (10)	28 (10)	5 (2)	6 (2)
Q5	I have poor appetite			208 (73)	208 (73)	54 (19)	54 (19)	22 (8)	22 (8)	0	0
Q6	I feel stuffy in my chest			214 (75)	214 (75)	46 (16)	42 (15)	20 (7)	18 (6)	4 (2)	10 (4)
Q7	I feel insecure in my life			130 (46)	130 (46)	104 (37)	104 (37)	40 (14)	40 (14)	10 (4)	10 (4)
Q8	I feel tired and weak			140 (49)	140 (49)	92 (32)	92 (32)	35 (12)	34 (12)	17 (6)	18 (6)
Q9	I feel loss of pleasure in normal activities			72 (25)	72 (25)	124 (44)	124 (44)	51 (18)	50 (18)	37 (13)	38 (13)
Q10	I do not remember things well			113 (40)	92 (32)	124 (44)	98 (35)	21 (7)	40 (14)	26 (9)	54 (19)
Q11	I feel difficulty concentrating			124 (44)	104 (37)	135 (48)	130 (46)	19 (7)	36 (13)	6 (2)	14 (5)
Q12	I feel slow in action and thinking			148 (52)	168 (59)	117 (41)	78 (27)	13 (5)	24 (8)	6 (2)	14 (5)
Q13	I fear of failure very much			142 (50)	142 (50)	82 (29)	82 (29)	37 (13)	36 (13)	23 (8)	24 (8)
Q14	I have no hope of my life			176 (62)	192 (68)	80 (28)	68 (24)	26 (9)	16 (6)	2 (1)	8 (3)
Q15	I think I am going to be sick			140 (49)	176 (62)	83 (29)	74 (26)	47 (17)	28 (10)	14 (5)	6 (2)
Q16	I observe that the worst is happening			192 (68)	140 (49)	68 (24)	68 (24)	16 (6)	48 (17)	8 (3)	28 (10)

* Self-rated measure for depressive prodrome after attending the jogging class; ** self-rated measure for depressive prodrome before attending the jogging class. ^†^ Cronbach’s alpha of 0.70 and above is good, 0.80 and above is better, and 0.90 and above is best.

**Table 2 ijerph-17-07705-t002:** The latent factors described the adolescent depressive prodrome via the related symptoms defined in BSM-5. The eigenvalue described the explanatory amount of variance of the integrated effect of the symptoms.

Symptom		R	R2	%	Eigenvalue (%)	Latent Factor of the Measure
Loss confidence	Q13	0.79	0.63	18%	6.16 (70%)	* AF1 (emotional comorbid)
Feeling **despondency**	Q2	0.77	0.60	17%		
Persistent sadness	Q1	0.73	0.53	15%		
Sickness	Q15	0.69	0.47	13%		
Loss pleasure	Q9	0.66	0.43	12%		
Pessimistic delusion	Q16	0.57	0.32	9%		
Feeling insecurity	Q7	0.55	0.31	9%		
Irritability	Q3	0.52	0.27	7%		
Risk of suicide	Q14	0.73	0.53	22%	1.33 (15%)	* AF2 (somatic comorbid)
Sleep disturbance	Q4	0.70	0.49	20%		
Poor appetite	Q5	0.70	0.48	20%		
Feeling fatigue	Q8	0.61	0.37	15%		
Focus deficit	Q11	0.75	0.57	23%		
Poor memory	Q10	0.73	0.54	50%	1.31 (15%)	* AF3 (cognitive comorbid)
Action retarding	Q12	0.72	0.52	50%		
Persistent sadness	Q1	0.76	0.58	18%	7.06 (74%)	** BF1 (emotional comorbid)
Feeling **despondency**	Q2	0.76	0.57	17%		
Irritability	Q3	0.56	0.32	10%		
Chest stuffy	Q6	0.52	0.27	8%		
Feeling insecurity	Q7	0.50	0.25	7%		
Loss pleasure	Q9	0.57	0.32	10%		
Loss confidence	Q13	0.74	0.54	16%		
Pessimistic delusion	Q16	0.67	0.45	14%		
Feeling fatigue	Q8	0.62	0.39	20%	1.34 (14%)	** BF2 (cognitive comorbid)
Poor memory	Q10	0.73	0.53	27%		
Focus deficit	Q11	0.76	0.57	29%		
Action retarding	Q12	0.69	0.48	24%		
Sleep disturbance	Q4	0.72	0.51	34%	1.15 (12%)	** BF3 (somatic comorbid)
Poor appetite	Q5	0.64	0.41	27%		
Sickness	Q15	0.77	0.60	39%		

* after attending the campus jogging class. ** before attending the campus jogging class.

**Table 3 ijerph-17-07705-t003:** List of three latent factors BF1, BF2, and BF3 analyzed before attending in the jogging class, and AF1, AF2, and AF3 after attending in the jogging class with the threshold score, false positive rate (FPR), sensitivity, false negative rate (FNR), predictive probability (PP), and the prevalence rate (PR) computed from binary logistic regression model for latent factors.

Threshold	Latent Factor	FPR	Sensitivity	FNR	PP	PR
Score 35	BF1	0.04	0.72	0.08	0.96	0.23
	BF2	0.29	0.63	0.11	0.71	
	BF3	0.1	0.28	0.17	0.9	
	BF1, BF2, BF3 covariate	0.07	0.81	0.05	0.99	
Score 36	BF1	0.17	0.74	0.06	0.83	0.19
	BF2	0.29	0.74	0.06	0.71	
	BF3	0.2	0.3	0.14	0.8	
	BF1, BF2, BF3 covariate	0.06	0.63	0.08	0.99	
Score 37	BF1	0.2	0.83	0.03	0.83	0.17
	BF2	0.32	0.79	0.04	0.68	
	BF3	0.4	0.25	0.14	0.6	
	BF1, BF2, BF3 covariate	0.2	0.83	0.03	--	
Score 38	BF1	0.21	0.95	0.01	0.8	0.14
	BF2	0.43	0.8	0.04	0.57	
	BF3	0.4	0.3	0.11	0.6	
	BF1, BF2, BF3 covariate	0	0.34	0.02	--	
Score 35	AF1	0.16	0.77	0.05	0.84	0.17
	AF2	0.19	0.44	0.1	0.81	
	AF3	0.24	0.4	0.1	0.76	
	AF1, AF2, AF3 covariate	0.22	0.9	0.02	0.99	
Score 36	AF1	0.3	0.78	0.04	0.7	0.14
	AF2	0.23	0.5	0.08	0.77	
	AF3	0.24	0.48	0.08	0.76	
	AF1, AF2, AF3 covariate	0	0.63	0.06	0.99	
Score 37	AF1	0.36	0.88	0.02	0.64	0.11
	AF2	0.38	0.5	0.06	0.61	
	AF3	0.28	0.56	0.05	0.72	
	AF1, AF2, AF3 covariate	0.04	0.75	0.03	0.99	
Score 38	AF1	0.39	0.96	0	0.6	0.1
	AF2	0.39	0.57	0.05	0.62	
	AF3	0.4	0.54	0.05	0.6	
	AF1, AF2, AF3 covariate	0.04	0.86	0.02	--	

**Table 4 ijerph-17-07705-t004:** Transformed proximities described by multidimensional scale (MS) before attending campus jogging program.

Before Attending Campus Jogging Program	BQ1	BQ2	BQ3	BQ4	BQ5	BQ6	BQ7	BQ8	BQ9	BQ10	BQ11	BQ12	BQ13	BQ14	BQ15	BQ16
BQ1	0.00															
BQ2	0.74	0.00														
BQ3	0.78	0.95	0.00													
BQ4	0.85	1.00	0.87	0.00												
BQ5	0.85	1.00	0.86	0.74	0.00											
BQ6	0.75	0.93	0.95	0.82	0.84	0.00										
BQ7	0.81	0.77	0.92	0.87	0.89	0.81	0.00									
BQ8	0.98	1.01	1.04	0.98	0.96	1.02	0.82	0.00								
BQ9	1.06	0.90	1.19	1.19	1.23	1.18	0.90	1.04	0.00							
BQ10	1.31	1.19	1.34	1.27	1.32	1.39	1.14	1.11	1.05	0.00						
BQ11	0.88	0.79	0.99	0.99	1.00	0.98	0.74	0.78	0.82	0.98	0.00					
BQ12	0.83	0.92	0.95	0.95	0.87	0.90	0.84	0.92	1.02	1.15	0.74	0.00				
BQ13	0.88	0.82	0.99	1.06	1.05	1.02	0.86	1.12	0.96	1.13	0.92	0.97	0.00			
BQ14	0.84	0.95	0.89	0.90	0.83	0.84	0.88	1.01	1.12	1.29	0.93	0.81	0.94	0.00		
BQ15	0.78	0.95	0.89	0.79	0.75	0.78	0.79	0.84	1.14	1.28	0.94	0.87	1.06	0.94	0.00	
BQ16	0.92	0.87	1.07	1.04	1.08	1.04	0.87	1.06	0.94	1.13	0.92	0.98	0.76	0.99	1.06	0.00

BQ# presented the symptom: 1 = crying spells with anxiety and stress, 2 = self-loathing of the sensitivity to pain and negative circumstances, 3 = feeling of agitation, 4 = not able to sleep well, 5 = appetite change, 6 = unexplained aches and pains, 7 = feeling nervousness, 8 = feeling fatigued, 9 = loss of pleasure doing daily activities, 10 = feeling the decline in memory, 11 = feeling short of attention, 12 = slowing down of thought, 13 = loss confidence, 14 = feeling hopeless, 15 = sickness with fever, vomiting or diarrhea, and 16 = having pessimistic delusion.

**Table 5 ijerph-17-07705-t005:** Transformed proximities described by multidimensional scale (MS) after attending campus jogging program.

After Attending Campus Jogging Program	AQ1	AQ2	AQ3	AQ4	AQ5	AQ6	AQ7	AQ8	AQ9	AQ10	AQ11	AQ12	AQ13	AQ14	AQ15	AQ16
AQ1	0.00															
AQ2	0.77	0.00														
AQ3	0.70	0.95	0.00													
AQ4	0.82	1.00	0.84	0.00												
AQ5	0.79	1.00	0.80	0.73	0.00											
AQ6	0.70	0.92	0.87	0.80	0.78	0.00										
AQ7	0.81	0.77	0.91	0.87	0.89	0.83	0.00									
AQ8	0.96	1.01	1.03	0.98	0.96	1.01	0.82	0.00								
AQ9	1.09	0.90	1.21	1.19	1.23	1.19	0.90	1.04	0.00							
AQ10	1.06	1.09	1.07	1.08	1.06	1.13	1.03	1.06	1.13	0.00						
AQ11	0.80	0.89	0.87	0.93	0.89	0.87	0.84	0.90	1.03	0.85	0.00					
AQ12	0.80	0.96	0.86	0.93	0.82	0.85	0.91	0.97	1.14	0.94	0.70	0.00				
AQ13	0.88	0.82	0.99	1.06	1.04	1.02	0.86	1.12	0.96	1.10	0.98	1.03	0.00			
AQ14	0.70	0.94	0.81	0.77	0.71	0.74	0.78	0.85	1.14	1.04	0.84	0.83	1.03	0.00		
AQ15	0.82	0.83	0.95	0.95	0.97	0.95	0.86	1.01	0.98	1.01	0.89	0.95	0.79	0.94	0.00	
AQ16	0.79	0.95	0.85	0.89	0.83	0.78	0.88	1.00	1.11	1.08	0.86	0.86	0.93	0.88	0.93	0.00

AQ# presented the symptom: 1 = crying spells with anxiety and stress, 2 = self-loathing of the sensitivity to pain and negative circumstances, 3 = feeling of agitation, 4 = not able to sleep well, 5 = appetite change, 6 = unexplained aches and pains, 7 = feeling nervousness, 8 = feeling fatigued, 9 = loss of pleasure doing daily activities, 10 = feeling the decline in memory, 11 = feeling short of attention, 12 = slowing down of thought, 13 = loss confidence, 14 = feeling hopeless, 15 = sickness with fever, vomiting or diarrhea, and 16 = having pessimistic delusion.

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
