# Peer review of "Secondary Prevention of Depressive Prodrome in Adolescents: Before and after Attending a Jogging Program on Campus"

_ijerph, 2020, doi:10.3390/ijerph17217705_

Round 1
Reviewer 1 Report
The authors present results from validating a novel measure of "prodromal depression", a construct they seem to have defined, and further describe results from a physical activity intervention on this construct in a sample of high school students.
I unfortunately cannot recommend this paper for publication. There are several reasons that the authors should address before submitting it to another journal.
The manuscript is generally poorly written and difficult to understand. Extensive English language editing is required.
The paper fails to provide information on whether participants gave informed consent, and whether the legal guardians of underage participants consented on their behalf.
The proposed construct of "prodromal depression" is not well described, and it is in my view unlikely that there is actually a prodrome of depression. The authors seem to transfer categories from the ultra-high risk of psychosis literature to depression, but this may not be justified. The authors fail to discuss the literature on the continuity of depressive symptoms between clinical and non-clinical levels, and the existing epidemiological literature on sub-threshold symptom levels as a risk factor for development of depression. I refer the authors to publications such as, which clearly should have been cited:
Solomon A, Haaga DAF, Arnow BA. Is clinical depression distinct from subthreshold depressive symptoms? A review of the continuity issue in depression research. J Nerv Ment Dis. 2001;189(8):498-506.
Crockett MA, Martínez V, Jiménez-Molina Á. Subthreshold depression in adolescence: Gender differences in prevalence, clinical features, and associated factors. J Affect Disord. 2020.
Klein DN, Shankman SA, Lewinsohn PM, Seeley JR. Subthreshold depressive disorder in adolescents: Predictors of escalation to full-syndrome depressive disorders. J Am Acad Child Adolesc Psychiatry. 2009;48(7):703-10.
They also incorrectly state that "So far to date, it remains unknown if any of school-related factors increase the risk for developing mental health problems and which factors are protective and help children and adolescents grow up mentally healthy", which clearly misrepresents the state of knowledge about risk and protective factors for development of adolescent depression.
Author Response
The authors present results from validating a novel measure of "prodromal depression", a construct they seem to have defined, and further describe results from a physical activity intervention on this construct in a sample of high school students.
Response to the reviewer:
Referred to the reference [Larson, M.K., E.F. Walker, and M.T. Compton, Early signs, diagnosis and therapeutics of the prodromal phase of schizophrenia and related psychotic disorders. Expert Rev Neurother, 2010. 10(8): p. 1347-59.], please check the general definition of depressive prodrome. If you disagree, please point out for us to modify it. The depressive prodrome is defined as the state when early signs and symptoms of the adolescent depression appear, but not yet clinically specific. In this study, we used a self-rated measure to check the severity status of the depressive prodrome before and after attending the jogging class on campus as a secondary prevention intervention for high school students in Taiwan.
I unfortunately cannot recommend this paper for publication. There are several reasons that the authors should address before submitting it to another journal.
Response to the reviewer:
Thank you for your suggestion.
The manuscript is generally poorly written and difficult to understand. Extensive English language editing is required.
Response to the reviewer:
Thank you for the suggestion, I’ll try my best to improve my English writing to you in the revision of this manuscript.
The paper fails to provide information on whether participants gave informed consent, and whether the legal guardians of underage participants consented on their behalf.
Response to the reviewer:
You can request a submitted copy of [informed consent] from the editorial office.
The proposed construct of "prodromal depression" is not well described, and it is in my view unlikely that there is actually a prodrome of depression.
Response to the reviewer:
From line 57 to 60 in our manuscript, we added a reference to the definition of depressive prodrome: the state when early signs and symptoms of the adolescent depression appear, but not yet clinically specific or severe as an adolescent depressive prodrome. We believe the occurrence of the actual depressive prodrome depends on the false positive rate of the measure. The depressive prodrome has been well defined in many references. The purpose of this study is to check the validity and reliability of our dimensional self-rated measure to assess the depressive prodrome via the FPR (false positive rate) and a logistic model. However, you may criticize our study from the point of view of the categorical approach. There is no right or wrong, just different points of view.
The authors seem to transfer categories from the ultra-high risk of psychosis literature to depression, but this may not be justified.
Response to the reviewer:
We did not use any literature of ultra-high risk of psychosis measure to evaluate the severity of adolescent depressive prodrome in our study. If you feel so, please specify the name of the literature so that we can modify it. Even so, it is generally believed that the depressive prodrome may involve signs of high-risk factors of psychosis. We referred to the definition from the article “Early signs, diagnosis, and therapeutics of the prodromal phase of schizophrenia and related psychotic disorders. Expert Rev Neurother. 2010 Aug; 10(8): 1347–1359. doi: 10.1586/ern.10.93”, which is a good article.
The authors fail to discuss the literature on the continuity of depressive symptoms between clinical and non-clinical levels, and the existing epidemiological literature on sub-threshold symptom levels as a risk factor for development of depression. I refer the authors to publications such as, which clearly should have been cited:
Response to the reviewer:
This study was not a clinical trial. It was only a dimensional approach of using a self-rated severity measure for high school students to express the probability of the severity status changing of the depressive prodrome before and after attending a jogging program. Meanwhile, the object of this study was to check the FPR effect on the prevalence of the depressive prodrome under the condition of specific SC values (severity score) in adolescents among high school students. The participants for this study were not depressive patients under any conditions. Therefore, we did not follow up on the continuity of depressive symptoms. If you disagree, please specify your point of view so that we can revise it. Your comments seem out of our topic of the research.
Solomon A, Haaga DAF, Arnow BA. Is clinical depression distinct from subthreshold depressive symptoms? A review of the continuity issue in depression research. J Nerv Ment Dis. 2001;189(8):498-506.
Crockett MA, Martínez V, Jiménez-Molina Á. Subthreshold depression in adolescence: Gender differences in prevalence, clinical features, and associated factors. J Affect Disord. 2020.
Klein DN, Shankman SA, Lewinsohn PM, Seeley JR. Subthreshold depressive disorder in adolescents: Predictors of escalation to full-syndrome depressive disorders. J Am Acad Child Adolesc Psychiatry. 2009;48(7):703-10.
Response to the reviewer:
Thanks for the suggestion. I read a lot of articles including these three articles several months ago, however, I didn’t think it would be necessary to cite those clinical articles, which are un-correlated with the adolescent depressive prodrome that we defined in this manuscript. However, based on your suggestion, I added a paragraph in results and discussion to specify the difference in the points of view.
They also incorrectly state that "So far to date, it remains unknown if any of school-related factors increase the risk for developing mental health problems and which factors are protective and help children and adolescents grow up mentally healthy", which clearly misrepresents the state of knowledge about risk and protective factors for development of adolescent depression.
Response to the reviewer:
This paragraph was cited from Reference [8]. If you think it is not correct, please provide your opinion specifically. We talked about the self-rated severity measure to assess the state status of the depressive prodrome of the students. However, you talked about the depression of the patients. These are two different issues in research, students are not patients. Anyway, thank you for your suggestion, I revised my manuscript and emphasized our research object more specific.
Reviewer 2 Report
Thank you for the opportunity to review this manuscript. I have a few minor comments for the authors to consider:
- The title may be revised to highlight the aim of the study, i.e. around the validity/false positive rate for adolescents. It is not necessary to specify Taiwanese in the title.
- The authors may include additional specific results in the abstract, provided it fits within the word limit.
- The authors may discuss the effect of ‘engagement’ on the validity. For example, the adolescents might have felt connected with the counsellors and the coach from the intervention. How would this translate to other interventions (likely to be translatable, albeit likelihood of selection bias from a convenience sampling of adolescents), and other non-interventions (perhaps less likely since other factors noted on line 45 are not explored entirely in the present study). The authors could add the implications and suggestions for future studies in the discussion section.
- It would be good to note SC = severe score on the footnote to Figure 1B.
Author Response
The title may be revised to highlight the aim of the study, i.e. around the validity/false positive rate for adolescents. It is not necessary to specify Taiwanese in the title.
Response to the reviewer:
Thank you for the suggestion. We'll change it to
Secondary Prevention of Depressive Prodrome in Adolescents Before and After Attending A Jogging Program as an Early Preventive Intervention
The authors may include additional specific results in the abstract, provided it fits within the word limit.
Response to the reviewer:
Thank you for the suggestion. We added some modifications to present our results in the abstract.
The authors may discuss the effect of ‘engagement’ on the validity. For example, the adolescents might have felt connected with the counsellors and the coach from the intervention. How would this translate to other interventions (likely to be translatable, albeit likelihood of selection bias from a convenience sampling of adolescents), and other non-interventions (perhaps less likely since other factors noted on line 45 are not explored entirely in the present study). The authors could add the implications and suggestions for future studies in the discussion section. It would be good to note SC = severe score on the footnote to Figure 1B.
Response to the reviewer:
Thank you for the suggestion. We added a paragraph based on your suggestion in the discussion of the manuscript.
Reviewer 3 Report
Obviously the topic of the paper is very important as it refers to depression in adolescents.
However, I would like to see the results of similar studies using the DSM-5 scale to facilitate comparison of results. The introduction could benefit from some additional information especially in adolescents.
Similarly the conclusions should expand a bit more and the authors can make some additional suggestions for further actions especially for the Taiwanese youth.
A thorough English check should be made as there are several parts that need rephrasing and use correct grammar as in lines 34-37, 44-46, 92-93, 260-261.
Also correct use of verbs must be observed e.g. lines 46, 48 we concern???
Author Response
However, I would like to see the results of similar studies using the DSM-5 scale to facilitate comparison of results. The introduction could benefit from some additional information especially in adolescents.
Response to the reviewer:
Thanks for your suggestion, we added more information about the depression in adolescent using DSM-5 scale in section III and compared them with our results in the conclusion.
Similarly the conclusions should expand a bit more and the authors can make some additional suggestions for further actions especially for the Taiwanese youth.
Response to the reviewer:
Thanks for the suggestion, we added a paragraph to present our planning for further studies in conclusion.
A thorough English check should be made as there are several parts that need rephrasing and use correct grammar as in lines 34-37, 44-46, 92-93, 260-261.
Response to the reviewer:
Thank you for your suggestion. Even if we could not find any serious syntax error, we made some changes to the semantics in lines 34-37, 44-46, 92-93, and 260-261.
Also correct use of verbs must be observed e.g. lines 46, 48 we concern???
Response to the reviewer:
Thank you for your suggestion, we made some changes to the semantics in lines 46-48 and checked all sentences in the manuscript.
The new version of the manuscript will be sent to you for review in a couple of days.
Round 2
Reviewer 1 Report
The authors have not adequately adressed the concerns raised in my previous review. It is very well that the editorial office has obtained the consent form, but the manuscript still fails to report on whether informed consent was obtained, as well as from whom and how.
The authors maintain their concept of prodromal depression, which is not well connected to the existing literature. The article they cite by Larsen, Walker and Compton concern psychotic disorders, not depression, and is simply not relevant to their data. Prodromal states is a well recognised (though recently somewhat controversial) concept in the study of psychotic disorders. The existence of a prodromal syndrome in depression is far less well established, and the authors further fail to properly review and cite the literature on it, such as:
Iacoviello, B. M., Alloy, L. B., Abramson, L. Y., & Choi, J. Y. (2010). The early course of depression: A longitudinal investigation of prodromal symptoms and their relation to the symptomatic course of depressive episodes. Journal of Abnormal Psychology, 119(3), 459–467.
Kouros, C. D., Morris, M. C., & Garber, J. (2016). Within-Person Changes in Individual Symptoms of Depression Predict Subsequent Depressive Episodes in Adolescents: a Prospective Study. Journal of abnormal child psychology, 44(3), 483-494.
What they have, as far as I can see, is some data from a non-randomised trial of aerobic exercise as a preventive measure for subthreshold adolescent depression. I would advise them to rework their conceptual framework and analysis completely in line with this. This should be a new submission entirely.
Sentences such as the following also demonstrate that further editing and professional language services are probably needed: "It took only 15 minutes for the participants to finish the self-rate measure of depressive prodrome assessment, but due to the high degree of the success of the assessment depends on advancing understanding of the prodrome [57, 58], we encouraged the participants consult questions in free with the professional counselors without keeping any personal record."
Author Response
The authors have not adequately adressed the concerns raised in my previous review. It is very well that the editorial office has obtained the consent form, but the manuscript still fails to report on whether informed consent was obtained, as well as from whom and how.
Response to the reviewer:
Thank you for the suggestion, I added, line21 to 21 and 126 to 127, about the informed consent obtained in the section of methods. Please check.
The authors maintain their concept of prodromal depression, which is not well connected to the existing literature. The article they cite by Larsen, Walker and Compton concern psychotic disorders, not depression, and is simply not relevant to their data. Prodromal states is a well recognised (though recently somewhat controversial) concept in the study of psychotic disorders. The existence of a prodromal syndrome in depression is far less well established, and the authors further fail to properly review and cite the literature on it, such as:
Respond to the reviewer:
We collected our data carefully. I disagree with the reviewer that our data is not related to depression. The Cronbach's Alpha of our questionnaire based on the depressive symptoms being defined by DSM-5 was 0.90, which was definitely reliable and valid. Please provide us your base so that we can explain the detail for you. There are very limited data on the prevalence and nature of prodrome of affective disorders, especially depressive disorders. For the purpose of the study of secondary prevention of adolescent depression, I believe either depressive prodrome or subthreshold depression should be a good approach. I also believe the dimensional approach is better than the categorical approach in secondary prevention. Therefore, the argument of the difference between depressive prodrome and subthreshold seems not necessary at all.
Iacoviello, B. M., Alloy, L. B., Abramson, L. Y., & Choi, J. Y. (2010). The early course of depression: A longitudinal investigation of prodromal symptoms and their relation to the symptomatic course of depressive episodes. Journal of Abnormal Psychology, 119(3), 459–467.
Respond to the reviewer:
Thank you for the suggestion. I learned from the article that you suggested to read. As I understand, the authors presented the longitudinal evidence for prodromal symptoms of depression episodes, which was an important concern for primary care. The authors also used a model to describe the relations between prodromal and residual symptoms. However, in our study, we are only concerned about the early sign of depressive symptoms for the purpose of high school campus secondary prevention in adolescents. The self-rated measure was proposed for the use of adolescent secondary prevention. We wanted to see if the early preventive intervention can affect the early sign of depressive symptoms. Therefore, we used the term of depressive prodrome to approach an integrated state of the depressive symptoms. That was why we didn’t cite this article at the time. In the revised version of our manuscript, I did the comparisons on the line from 32 to 37 and 116 to 121.
Kouros, C. D., Morris, M. C., & Garber, J. (2016). Within-Person Changes in Individual Symptoms of Depression Predict Subsequent Depressive Episodes in Adolescents: a Prospective Study. Journal of abnormal child psychology, 44(3), 483-494.
Respond to the reviewer:
Thank you for the suggestion. I learned from the article that you suggested to read. As I understand, the topic of this article talked about was far from what we discussed in our study. Major Depressive Episode (MDE) in children is the concern for primary care in the clinic. The authors used the construct of Children’s Depression Rating Scale-Revised (CDRS-R) to figure out the depression episode for the children. The children's gender effect was specified in their study, which was not secondary prevention-related at all.
What they have, as far as I can see, is some data from a non-randomised trial of aerobic exercise as a preventive measure for subthreshold adolescent depression. I would advise them to rework their conceptual framework and analysis completely in line with this. This should be a new submission entirely.
Thank you for your suggestion.
Sentences such as the following also demonstrate that further editing and professional language services are probably needed: "It took only 15 minutes for the participants to finish the self-rate measure of depressive prodrome assessment, but due to the high degree of the success of the assessment depends on advancing understanding of the prodrome [57, 58], we encouraged the participants consult questions in free with the professional counselors without keeping any personal record."
Respond to the reviewer:
Thank you for your suggestion. I polished my English sentence by sentence and rewrite some parts of the paragraphs.